# Shifts in evolutionary lability underlie independent gains and losses of root-nodule symbiosis in a single clade of plants

Heather R. Kates [1] ✉, Brian C. O'Meara [2], Raphael LaFrance[1], Gregory W. Stull [3], Euan K. James [4], Shui-Yin Liu[3], Qin Tian [3], Ting-Shuang Yi [3], Daniel Conde [5], Matias Kirst [6,7], Jean-Michel Ané [8,9], Douglas E. Soltis [1,6,10,11], Robert P. Guralnick [1,10], Pamela S. Soltis [1,6,10] & Ryan A. Folk [12] ✉

Root nodule symbiosis (RNS) is a complex trait that enables plants to access atmospheric nitrogen converted into usable forms through a mutualistic relationship with soil bacteria. Pinpointing the evolutionary origins of RNS is critical for understanding its genetic basis, but building this evolutionary context is complicated by data limitations and the intermittent presence of RNS in a single clade of ca. 30,000 species of flowering plants, i.e., the nitrogen-fixing clade (NFC). We developed the most extensive de novo phylogeny for the NFC and an RNS trait database to reconstruct the evolution of RNS. Our analysis identifies evolutionary rate heterogeneity associated with a two-step process: An ancestral precursor state transitioned to a more labile state from which RNS was rapidly gained at multiple points in the NFC. We illustrate how a two-step process could explain multiple independent gains and losses of RNS, contrary to recent hypotheses suggesting one gain and numerous losses, and suggest a broader phylogenetic and genetic scope may be required for genome-phenome mapping.

The origin of complex traits is among the most compelling problems in evolutionary biology[1]. Molecular and genetic evidence has demonstrated that novel complex traits often originate by repurposing existing molecular machinery, leading to a more nuanced multi-level view of trait homology[2–5]. Such molecular repurposing provides a mechanistic explanation for how complex traits, seemingly "difficult" to evolve, are convergently gained multiple times[6–9]. Clarifying the evolutionary processes underlying phenotypes that enable symbiotic partnerships is particularly challenging, because symbiotic traits facilitate a coordinated dance among prospective partners[10].

Root nodule symbiosis (RNS) between angiosperms and their nitrogen-fixing bacterial symbionts is one of the most ecologically significant symbiotic traits found in nature. How RNS initially arose

[1]Florida Museum of Natural History, University of Florida, Gainesville, FL, USA. [2]Department of Ecology and Evolutionary Biology, University of Tennessee, Knoxville, TN 37996-1610, USA. [3]Germplasm Bank of Wild Species, Kunming Institute of Botany, Chinese Academy of Sciences, Kunming 650201 Yunnan, China. [4]The James Hutton Institute, Invergowrie Dundee, Scotland, UK. [5]Centro de Biotecnología y Genómica de Plantas (CBGP), Universidad Politécnica de Madrid (UPM)-Instituto Nacional de Investigación y Tecnología Agraria y Alimentaria (INIA-CSIC), Campus de Montegancedo, Pozuelo de Alarcón, Madrid 28223, Spain. [6]Genetics Institute, University of Florida, Gainesville, FL, USA. [7]School of Forest, Fisheries and Geomatic Sciences, University of Florida, Gainesville, FL, USA. [8]Department of Bacteriology, University of Wisconsin-Madison, Madison, WI 53706, USA. [9]Department of Agronomy, University of Wisconsin-Madison, Madison, WI 53706, USA. [10]Biodiversity Institute, University of Florida, Gainesville, FL, USA. [11]Department of Biology, University of Florida, Gainesville, FL, USA. [12]Department of Biological Sciences, Mississippi State University, Mississippi State, MS, USA. ✉e-mail: hkates@ufl.edu; rfolk@biology.msstate.edu

in relation to the genomic toolkit underlying this trait and how many times this symbiotic relationship has been gained or lost remains uncertain and contentious[11–14]. Continued debate on the number and distribution of gains and losses of RNS is primarily due to the highly intermittent presence of this trait across a large angiosperm clade of ca. 30,000 species, the nitrogen-fixing clade (NFC)[15–17].

Root nodule symbiosis is sporadically distributed across four diverse subclades of the NFC (Fabales, Rosales, Fagales, Cucurbitales) and occurs with two different bacterial symbionts—rhizobia (Alpha- and Betaproteobacteria) in Fabales and *Parasponia* (Rosales: Cannabaceae)[18] and *Frankia* (Actinobacteria) in all other RNS taxa (i.e., the actinorhizal lineages[19]). Such a pattern could have resulted from a single origin of the symbiosis with a large number of subsequent losses[11], from multiple gains of diverse symbioses that share a deeper homology[12,15,20], or from some combination of gains and losses[15]. Identifying which of these histories underlies the diversity and phylogenetic distribution of RNS provides an evolutionary framework to test hypotheses related to how nodulation evolved[21] and has broad agronomic implications for ongoing efforts to induce nodulation in non-nodulating food crops, particularly cereals. The simplest hypothesis, a single origin of the RNS trait by coopting symbiosis pathways that occur broadly in plants[11,22], is currently widely accepted among functional and molecular biologists and would suggest genetic homology across species within and outside the NFC subclades, indicating that candidate species for genetic transformation already possess some essential genes. This hypothesis suggests a simpler transformation might feasibly induce symbiotic nitrogen fixation in non-RNS lineages[23]. However, a single-origin, multiple-loss history of the trait could also imply that a limited evolutionary palette is available to inform genetic transformation. Under the alternative hypothesis of multiple independent gains, where nodulation in different NFC subclades is non-homologous, multiple pathways to nodulation are likely available for genetic transformation. At the same time, more limited homology would also suggest a greater degree of genetic novelty among phylogenetically diverse nodulators[13] and, therefore, pose further challenges to inducing competency for nodulation in candidate species[23,24].

Determining the underlying deep homology of complex, multi-level traits requires a densely and proportionally sampled phylogenetic framework for ancestral character state reconstruction. Although this requirement has long been acknowledged[13], methods to assemble and analyze a dataset of thousands of species are only recently available[25,26]. Given the complex nature of RNS and its sporadic distribution in the NFC, dense taxon sampling and informative phylogenetic resolution are imperative. However, current considerations of RNS evolution still rely on phylogenies with limited taxon and trait sampling[13]. While fully resolved phylogenies remain elusive for many deep branches in the plant tree of life such as those in the NFC[27,28], results based on large, multi-locus nuclear datasets offer clear advantages over those inferred from other types of data[29]. We construct such a phylogeny of the NFC based on ~13,000 species to ask whether RNS evolved once or many times and identify the origins of RNS within the NFC.

## Results and discussion

### The ancestor of the nitrogen-fixing clade lacked nodular nitrogen-fixing symbiosis

We used dense sampling of herbarium specimens as the source of material for sequencing a panel of 100 single- or low-copy nuclear loci[26] to infer the phylogeny of the NFC. The result is, to our knowledge, the single largest de novo phylogenetic dataset constructed to date (n = 12,768 ingroup species; Fig. 1; Supplementary Fig. 1; Supplementary Data 1; Supplementary Table 1). After time-calibrating the NFC phylogeny based on a previous estimate for the larger rosid clade which includes the NFC[30] we replaced the NFC clade in that rosid

phylogeny with our dated tree. We then used a novel and updated RNS trait database (Supplementary Fig. 1) and the full phylogeny to reconstruct the history of the RNS trait and to model its transition rate (i.e., how "easy" the trait was to gain or lose through evolutionary history). To do this, we used hidden Markov models (HMMs) to test the fit of different evolutionary transition rate models (Supplementary Table 2) and then used the joint reconstruction of ancestral character states[31] to infer the history of the RNS trait and its varying lability based on the best-fitting model. (The combination of observed and reconstructed presence or absence of the RNS trait and the inferred evolutionary rate of transitions between different RNS traits is hereafter referred to as a "hidden state" or "state").

Our results indicate that the most recent common ancestor of the NFC lacked RNS; moreover, there was not a single origin of the symbiosis, but rather 16 independent gains and ten subsequent losses of RNS. We also constructed a phylogeny based on our complete sampling but constrained to an alternative backbone topology resolved in some previously published phylogenies[28,32] and found results similar to our best-evidence phylogenetic hypothesis (Supplementary Note 4; Supplementary Fig. 2; Supplementary Table 2). Thus, our findings are robust to different backbone topologies for the NFC.

The prevalence of single or multiple origin(s) of complex traits across the Tree of Life is currently debated[33]. These debates often center on mechanisms that underlie character identity[34] and quality of sampling. Both are relevant here, and our results suggesting multiple gains of RNS do not necessarily conflict with recent findings that critical functional genes, present in at least some NFC outgroups but without nodulation function, were recruited to form a genetic ensemble only once in the ancestor of the NFC[35]. However, this single assembly of what is currently recognized as the requisite genetic machinery need not coincide with a gain of RNS, which appears to require an additional "switch" that was itself gained multiple times and is instead emblematic of limited trait homology. Similar examples of "multi-level convergence", in which a phenotype arises via evolutionarily independent nested shifts, potentially at multiple levels of biological organization, are increasing as new data become available[36]. Further study – with expanded sampling beyond that of current reports[35] – is needed to identify the exact node(s) at which those genes currently considered fundamental to RNS were assembled.

Deep homology involving the convergent recruitment of genes has been found in C$_4$ photosynthesis[37], in betalain biosynthesis in the plant order Caryophyllales[38], in the adaptation of birds to high-elevation environments[39], and in evolutionary transitions from variously colored to red flowers[40], among others. The lack of fully homologous origins of RNS is unsurprising in terms of the trait's morphological diversity. Nodules in different NFC lineages are structurally and anatomically distinct from one another[41], and they recruit distantly related bacterial partners. A deeper understanding of both the evolutionary history of nodulation and the genetics underlying symbiotic states across lineages with diverse nodule types is needed to determine to what extent nodules across the NFC may have a similar underlying developmental and genetic basis[20].

### A high number of independent gains of RNS, followed by occasional loss

To quantify the number of independent gains of RNS, we counted transitions between nodes estimated in a non-nodulating state and those estimated in a nodulating state by joint estimation of ancestral character states[31]. Our results demonstrate 16 likely gains of RNS in the NFC, including six gains of rhizobial RNS in legumes and one in *Parasponia*, and nine gains of actinorhizal associations in the other families with RNS (Table 1). Such a high number of independent gains of a complex trait is not unprecedented; our results add to many examples illustrating that complex traits can evolve many times[7–9,33].

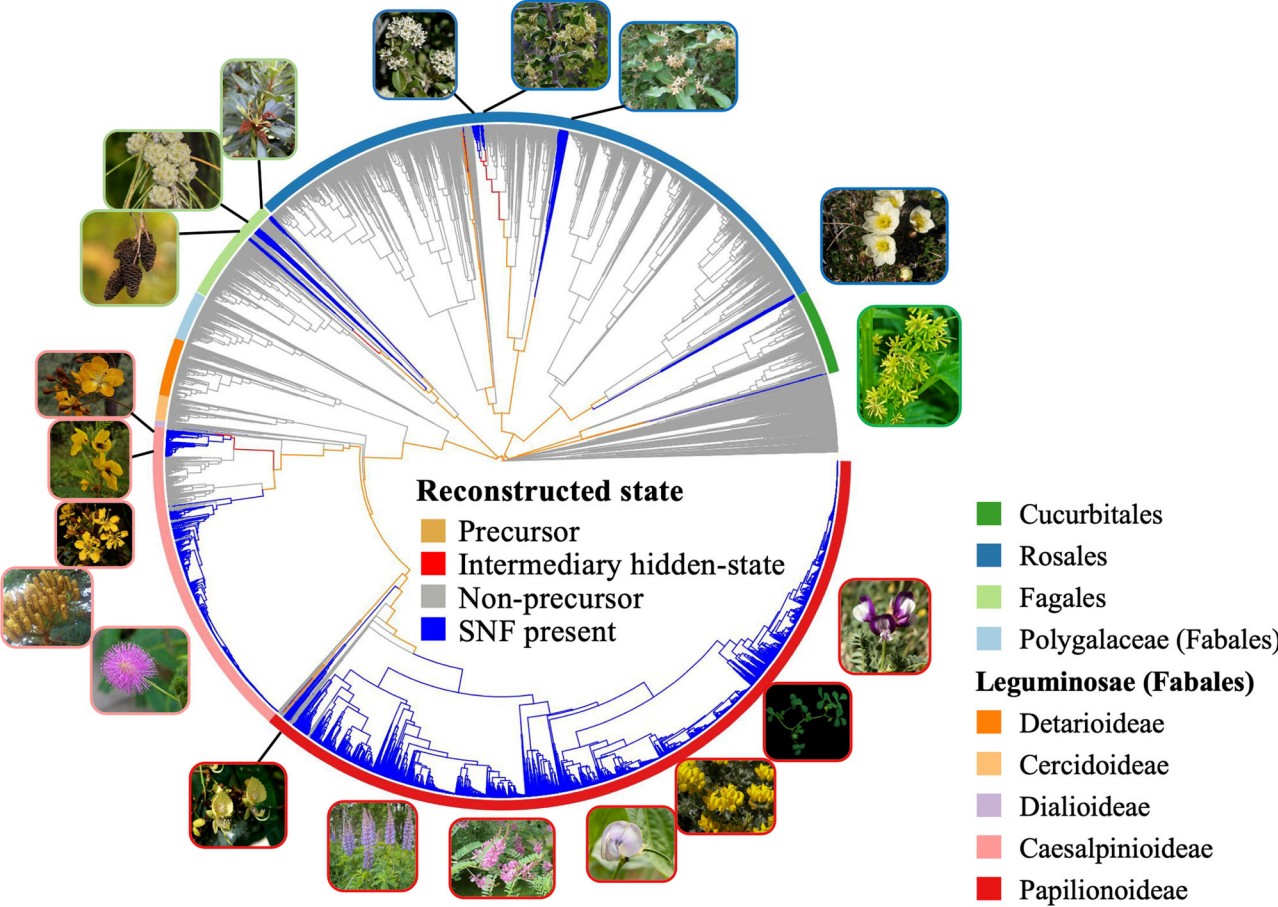

**Reconstructed state**
- Precursor
- Intermediary hidden-state
- Non-precursor
- SNF present

- Cucurbitales
- Rosales
- Fagales
- Polygalaceae (Fabales)

**Leguminosae (Fabales)**
- Detarioideae
- Cercidoideae
- Dialioideae
- Caesalpinioideae
- Papilionoideae

**Fig. 1 | Phylogeny and ancestral character state reconstruction of the nitrogen-fixing clade (NFC) in context within rosids.** NFC orders and legume subfamilies are indicated by colored bars; non-NFC clades are reduced in size (unlabeled section). Branches are colored by the state estimated at their tipward node, but to focus on the history of RNS-gain, all three hidden states of RNS-absent are colored blue. For each of 16 hypothesized independent gains, an image of a single representative taxon is indicated at its approximate phylogenetic position, except in the very species-rich clades where multiple representatives sharing a border color represent a single origin of RNS. Clockwise beginning at the top right in Papilionoideae, the pictured representatives are: (1) *Astragalus*, (2) *Medicago*, (3) *Phaseolus*, (4) *Indigofera*, (5) *Lupinus*, (6) *Swartzia*, (7) *Mimosa*, (8) *Dimorphandra*, (9) *Moldenhawera*, (10) *Chamaecrista*, (11) *Melanoxylum*, (12) *Alnus*, (13) *Casuarina*, (14) *Myrica*, (15) *Ceanothus*, (16) *Trevoa*, (17) *Elaeagnus*, (18) *Dryas, and* (19) *Datisca*.
Image credits: (1) *Astragalus* – Photo taken from Wikimedia user Kaldari from Wikipedia.org https://en.wikipedia.org/wiki/Astragalus_nuttallianus#/media/File: Fabaceae_flowers_Texas.jpg 2009. Public Domain. (2) *Medicago* – Photo taken from Wikimedia user Ninjatacoshell from Wikipedia.org https://en.wikipedia.org/wiki/Medicago_truncatula#/media/File:Medicago_truncatula_A17_branch.JPG 2009. CC BY-SA 3.0. (3) *Lotus* – Photo taken from Wikimedia user Hans Hillewaert from Wikipedia.org https://commons.wikimedia.org/wiki/Lotus_cytisoides#/media/File: Lotus_cytisoides.jpg 2008. CC BY-SA 3.0. (4) *Indigofera* – Photo taken from Wikimedia user Kurt Stüber from Wikipedia.org https://en.wikipedia.org/wiki/Indigofera_tinctoria#/media/File:Indigofera_tinctoria1.jpg 2004. CC BY-SA 3.0. (5) *Lupinus* – Photo taken from Wikimedia user Banana patrol from Wikipedia.org https://commons.wikimedia.org/wiki/File:Lupinus_polyphyllus.JPG 2005. CC BY-SA 3.0. (6) *Swartzia* - Photo taken from Wikimedia user Vojtěch Zavadil from Wikipedia.org https://commons.wikimedia.org/wiki/File:13010-Swartzia-picta-Caura.JPG 2007. CC BY-SA 3.0. (7) *Mimosa* - Photo taken from Wikimedia user Don McCulley from Wikipedia.org https://commons.wikimedia.org/wiki/Category:Mimosa_pudica_(flowers)#/media/File:Mimosa_pudica_IMG_0230.jpg 2018. CC BY-SA 4.0. (8) *Dimorphandra* - Photo taken from Wikimedia user Denis A. C. Conrado from Wikipedia.org. https://commons.wikimedia.org/wiki/File:Favadanta03.jpg 2007.

Use permitted by copyright holder. (9) *Moldenhawera* - Permission by Domingos Cardoso. (10) *Chamaecrista* - Photo taken from Wikimedia user Fritz Flohr Reynolds from Wikipedia.org https://commons.wikimedia.org/wiki/Category:Chamaecrista_fasciculata#/media/File:Chamaecrista_fasciculata_-_Partridge_Pea.jpg 2013. CC BY-SA 3.0. (11) *Melanoxylum* - Permission by Domingos Cardoso. (12) *Alnus* - Photo taken from Wikimedia user Noël Zia Lee from Wikipedia.org https://commons.wikimedia.org/wiki/Alnus#/media/File:Red_Alder_Female_Catkins_in_Autumn.jpg 2007. CC BY 2.0. (13) *Casuarina* - Photo taken from Wikimedia user Sam Fraser-Smith from Wikipedia.org https://commons.wikimedia.org/wiki/File:Casuarina_equisetifolia_L._-_Australian_pine,_beach_sheoak,_common_ironwood_(3771046132).jpg 2009. CC BY 2.0. (14) *Myrica* - Photo taken from Wikimedia user Ettrig from Wikipedia.org https://commons.wikimedia.org/wiki/File:Myrica_faya.jpg 2006. Public Domain. (15) *Ceanothus* - Photo taken from Wikimedia user Stan Shebs from Wikipedia.org https://en.wikipedia.org/wiki/Ceanothus_pauciflorus#/media/File:Ceanothus_greggii_4.jpg 2006. CC BY-SA 3.0. (16) *Trevoa* - Photo taken from Wikimedia user Dick Culbert from Wikipedia.org https://commons.wikimedia.org/wiki/Category:Trevoa_quinquenervia#/media/File:Trevoa_quinquenervia_of_the_Rhamnaceae_(8405983258).jpg 2006. CC BY 2.0 (17) *Elaeagnus* - Photo taken from Wikimedia user KENPEI from Wikipedia.org https://en.m.wikipedia.org/wiki/File:Elaeagnus_umbellata1.jpg 2008. CC BY-SA 2.1 (18) *Dryas* - Photo taken from Wikimedia user Kim Hansen from Wikipedia.org https://commons.wikimedia.org/wiki/File:Dryas_integrifolia_upernavik_2007_06_28_1.jpg 2007. CC BY-SA 3.0 (19) *Datisca* - Photo taken from Wikimedia user H. Zell from Wikipedia.org https://commons.wikimedia.org/wiki/Category:Datisca_cannabina#/media/File:Datisca_cannabina_002.JPG 2009. CC BY 3.0 Links to licenses for reuse restrictions: CC BY-SA 4.0: https://creativecommons.org/licenses/by-sa/4.0/ CC BY 3.0: https://creativecommons.org/licenses/by/3.0/deed.en CC BY-SA 2.1: https://creativecommons.org/licenses/by-sa/2.1/ca/deed.en CC BY 2.0: https://creativecommons.org/licenses/by/2.0/deed.en CC BY-SA 3.0: https://creativecommons.org/licenses/by-sa/3.0/deed.en Public Domain: https://wiki.creativecommons.org/wiki/public_domain.

**Table 1 | List of hypothesized origins of RNS in the NFC**

| Order | Family or Subfamily (Fabales) | Nodulating clade | Comparison | SI annotation |
|---|---|---|---|---|
| Fabales | Papilionoideae | meso-Papilionoideae (50 kb inversion clade)* | non-*Nissolia* Adesmia clade (+) and *Nissolia* (–)† | Sup.Figs. 7; 1 |
| Fabales | Papilionoideae | Atelioids and Swartzioids *sensu stricto** | Atelioids (+) and *Bocoa, Trischidium* (–)† | Sup.Figs. 7; 2 |
| Fabales | Papilionoideae | *Dussia* | Amburaneae exclusive of *Dussia* | Sup.Figs. 7; 3 |
| Fabales | Caesalpinioideae | Mimosoid Clade + Tachigali Clade + Dimorphandra Clade + Peltophorum group + *Moldenhawera** | *Anadenanthera*(+) and *Parkia* (–)†; *Viguier-anthus* (+?) and *Zapoteca* (–)† | Sup. Figs. 3; 4 |
| Fabales | Caesalpinioideae | *Chamaecrista* | Cassieae Clade or *Vouacapoua*†,§ | Sup. Figs. 3; 5 |
| Fabales | Caesalpinioideae | *Melanoxylum* + *Recordoxylon* | Cassieae Clade or *Vouacapoua*†,§ | Sup. Figs. 3; 6 |
| Rosales | Rosaceae | Subfamily Dryadoideae | Rosaceae exclusive of Dryadoideae | Sup. Figs.4;7 |
| Rosales | Cannabaceae | *Parasponia* | *Trema* (paraphyletic) | Sup. Figs.4;8 |
| Rosales | Rhamnaceae | *Ceanothus* | *Colubrina* (paraphyletic)† | Sup. Figs.4;9 |
| Rosales | Rhamnaceae | Tribe Colletieae | *Ziziphus* (paraphyletic)† | Sup. Figs.4;10 |
| Rosales | Elaeagnaceae | Elaeagnaceae | Dirachmaceae | Sup. Figs.6;11 |
| Fagales | Betulaceae | *Alnus* | Subfamily Coryloideae† | Sup. Figs.6;12 |
| Fagales | Casuarinaceae | Casuarinaceae | Ticodendraceae + Betulaceae | Sup. Figs.6;13 |
| Fagales | Myricaceae | Myricaceae exclusive of *Canacomyrica* | *Canacomyrica* | Sup. Figs.6;14 |
| Cucurbitales | Datiscaceae | *Datisca* (monogeneric) | Tetramelaceae | Sup. Figs.6;15 |
| Cucurbitales | Coriariaceae | *Coriaria* (monogeneric) | Corynocarpaceae | Sup. Figs.6;16 |

Clades that also include non-nodulating taxa due to loss of RNS are marked with *. Incongruences with previously published phylogenies are marked †. Comparison groups that are sister taxa with an ancestral absence of RNS are marked † except in those cases where they are taxa with secondary loss and their closest RNS-present relatives. §*Vouacapoua* is generally regarded as non-nodulating, but see Moreira (1992)[93].

$C_4$ photosynthesis likely evolved over 60 times in angiosperms[42], with 23 gains just in Caryophyllales ( ~ 12,000 species) alone[43]. The primary mechanism of $C_4$ gene evolution by gene duplication, non-functionalization, and neofunctionalization[44] established the importance of duplicating preexisting genetic material in generating remarkable trait lability. Despite intensive work to understand the underlying mechanisms of RNS in model legume species, with important results[45–48], little is known about how RNS is lost and gained, although polyploidy may have played a role, at least in the case of papilionoid legume nodules[49,50]. Even less is known about the mechanisms controlling RNS outside of legumes.

In addition to multiple gains, we also infer eight independent losses of RNS in Caesalpinioideae and two likely losses in Papilionoideae (Table 2). Based on our well-sampled and strongly supported topology and reconstruction, RNS loss is only observed in legumes; why the loss of RNS has not occurred in other subclades is a compelling question, especially given that our understanding of the genetic basis of RNS is based on legume systems. Most of these inferred losses occurred following a single gain in Caesalpinioideae, suggesting that nodules derived from that gain event may be of a type that is more easily lost than other nodular forms[51]. The overall pattern of gain and loss resembles that recovered previously[14], with more losses and gains reported here due in large part to our improved sampling of clades with high rates of RNS transition. Identifying the historical constraints and selective forces responsible for maintaining nodulation in natural systems would be a major step towards resolving such disparities in the evolutionary fate of symbiosis in different clades.

**Evolutionary rate shifts explain the phylogenetic pattern of RNS**
Because transition rates in gains and losses of characters can vary during evolution, we used hidden Markov models as implemented in the R package corHMM[52] to characterize how the evolutionary rates of nodulation loss and gain vary across the NFC. We find that the best model of RNS evolution is one in which there are three hidden states for non-nodulating lineages (Supplementary Table 2) that differ with respect to their pathway to gain the RNS phenotype. Lineages in a

"precursor" state can transition to an "intermediary hidden state" from which RNS is gained, and those in a "non-precursor" state cannot leave that state and thus have no pathway to gain nodulation (Fig. 2). Given the age and high species diversity of the rosid clade, the better model fit of the three-rate model compared to the two-rate classical precursor model[53] is unsurprising[52]. We also performed reconstructions of the evolution of RNS under this two-rate and other poorer-fit models that represent a range of evolutionary rate regimes including previously proposed models of RNS gain-loss (Supplementary Table 2) to address concerns that hidden-state model selection cannot directly account for external information about the origins of RNS. We found that the evolution of RNS described above is also reconstructed under these poorer-fit models (Supplementary Data Figs. 11 and 12), indicating that the reconstruction of RNS gain-loss is not sensitive to model choice and is consistent under previously proposed models.

Under our best-fit model, the most recent common ancestor of the NFC is in the precursor state (Fig. 1). This result again confirms that a change or set of changes shared by the NFC relative to the rest of all angiosperms likely explains the phylogenetic bias of RNS gains in angiosperms[14,15]. However, rather than localizing this precursor state at the ancestor of the NFC clade, we find a slightly earlier origin as part of the earliest rapid diversification of the large rosid lineage, of which the NFC is just one subclade[54], with no other extant rosid group maintaining the precursor. Our results indicate that lineages do not gain RNS directly from this precursor state—they first transition into an intermediary hidden state from which RNS is gained. This two-step pathway to gain RNS may indicate that, following the initial, shared predisposition to RNS present in the most recent common ancestor of the NFC, additional changes were required to catalyze the formation of the RNS phenotype.

Identifying an intermediary hidden state between precursor and RNS gain means that each independent gain of RNS is preceded by an evolutionary rate-shift distinct from that which characterizes the NFC more broadly. This strongly suggests that whatever change occurred in the most recent common ancestor of the NFC was necessary but not sufficient for the evolution of the RNS phenotype. RNS gain from the intermediary hidden state is 10 times faster than is a transition from

**Table 2 | List of hypothesized losses of RNS in the NFC**

| Order | Family or Subfamily (Fabales) | Loss clade | Comparison | SI annotation |
|---|---|---|---|---|
| Fabales | Papilionoideae | *Bocoa, Trischidium* | Atelioids | Sup.Figs. 7; 17 |
| Fabales | Papilionoideae | Nissolia | Adesmia clade exclusive of *Nissolia* | Sup.Figs. 7; 18 |
| Fabales | Caesalpinioideae | *Zapoteca* | *Viguieranthus* | Sup.Figs. 3; 19 |
| Fabales | Caesalpinioideae | *Vouacapoua* | *Melanoxylum + Recordoxylon* | Sup.Figs. 3; 20 |
| Fabales | Caesalpinioideae | *Parkia* | *Anadenanthera* | Sup.Figs. 3; 21 |
| Fabales | Caesalpinioideae | *Newtonia* | The most inclusive clade that includes *Prosopis* and *Inga* but not *Newtonia* | Sup.Figs. 3; 22 |
| Fabales | Caesalpinioideae | *Adenanthera, Tetrapleura* | RNS+ *Adenanthera* group | Sup.Figs. 3; 23 |
| Fabales | Caesalpinioideae | *Mora* | *Dimorphandra** | Sup.Figs. 3; 24 |
| Fabales | Caesalpinioideae | *Arapatiella* | *Jacqueshuberia* | Sup.Figs. 3; 25 |
| Fabales | Caesalpinioideae | *Peltophorum* group | The most inclusive clade that includes *Diptychandra* and *Inga* but not *Peltophorum* | Sup.Figs. 3; 26 |

The comparison is "none" when the loss clade is sister to a species-rich clade that includes independent gain and loss events. *See Supplementary Note 3.

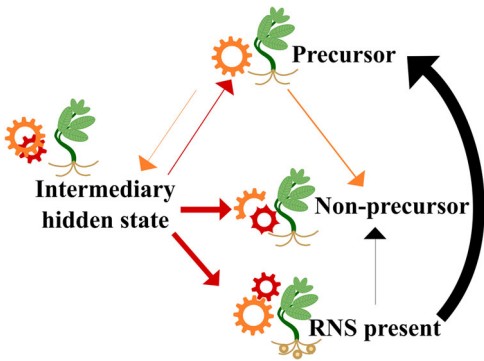

**Fig. 2 | Model of a two-step pathway to the gain of nodulation.** A simplified depiction of the rate transition network inferred by the best hidden rates model to highlight pathways to gaining RNS ("RNS-present") within the NFC. The three RNS-present hidden-states are combined into a single RNS-present (any rate category) state. Transition rates from each of the three inferred RNS-absent hidden states are drawn with width relative to the speed of the transition. Transition arrows are colored by the state from which they originate to correspond with how inferred states are colored on the phylogeny in Fig. 1. Two types of gears in the figure (orange and red) represent two levels of homology as predicted by a two-step pathway, only one of which occurs in the precursor. Subsequent changes in gear shape represent pseudogenization and other non-functionalization processes in nodulation pathways. An illustration of the full transition rate matrix is shown in Supplementary Fig. 13.

the precursor to the intermediary hidden state (Fig. 2). This rate difference suggests that the second step in the pathway to gaining RNS increases the lability of RNS evolution more dramatically than the initial predisposition. Except in the Rosales family Rhamnaceae and legume subfamily Caesalpinioideae, where multiple phylogenetically proximate gains of RNS share an origin of the intermediary hidden state (Fig. 1), each origin of RNS is descended from an independent intermediary hidden state (although we do not see this highly transient state mapped onto nodes by ancestral trait reconstruction due to its lability; Fig. 2).

Loss of RNS through a transition to a non-precursor state, an irreversible change, occurs at a rate 14 times slower than the rate at which the precursor transitions to the intermediary hidden state and 150 times slower than the rate at which RNS is then gained from the intermediary hidden state. Together, these transition rates characterize an evolutionary history of RNS loss and gain in which the RNS trait itself is much more readily gained than lost. However, *lability* is more likely lost than gained: the precursor state is more likely to transition to non-precursor than it is to transition to the intermediary hidden-state, and the intermediary hidden-state is equally likely to lose lability as it is to gain RNS. Taken together, these results suggest that elements of the genetic toolkit that may underlie the potential to gain RNS are easily lost, but RNS itself, once gained, is an evolutionarily stable strategy.

**Precise phylogenetic origins of RNS**

Determining where nodulation evolved in the NFC allows for identification of homology of nodule and symbiotic states within the clade. We have identified multiple gains of nodulation in each of the four orders in the NFC: two in Cucurbitales, five in Rosales, three in Fagales, and six in Fabales (Fig. 1; Table 1; Supplementary Figs. 3–6); some of these multiple gains yielded morphologically similar nodules, but others led to disparate nodule types, a finding consistent with limited homology of RNS among extant taxa[55]. Species representing each gain should be included in efforts to understand the underlying molecular mechanisms of RNS, or at least interpretational limits should be considered when entire clades are excluded from models of nodulation.

The many nodulating legume species that are well studied at the phenotypic and genetic levels represent a single homologous and relatively homogeneous nodulation trait in the Meso-Papilionoideae, a clade that includes the majority of papilionoid legumes[55]. This clade includes nearly all major legume crops and the model systems *Lotus japonicus* and *Medicago truncatula*. Our data indicate that the origin of RNS in this clade is independent from two other gains of RNS in Papilionoideae clades (Supplementary Fig. 7) in which nodulation traits and key regulatory genetic networks are not as well described, so that generalizability across the legumes would require representatives of these additional gains.

Any detailed estimate of the number and phylogenetic position of gain and loss events in evolutionary history is contingent on sampling. Here, the sampling strategy was purposefully focused on lineage representation, especially in those areas of the tree where nodulation states are most unstable based on our curated nodulation database (see Methods). This approach leads to an estimate that is likely robust to the impact of remaining missing species, many of which are in clades that are invariant in nodulation states based on current knowledge. More important than phylogenetic sampling is currently incomplete and biased knowledge of nodulation states, an area of crucial importance that is the subject of active research by other investigators. Our work provides a strong basis for better delimiting key lineages where nodulation states have been subject to rapid evolution. Further detailed study is needed on species from these under sampled parts of the tree, many of which were not previously of research interest due to poor phylogenetic resolution. Focused investigations resting on our firm phylogenetic framework may

generate new insights into the evolution of this symbiosis, including a possible revision of the number of origins of nodulation.

Overall, our broad-scale approach has resulted in a revised picture of the evolution of RNS. Based on our reconstructions, there are fewer losses than gains of RNS in the NFC, and these losses occur only in legumes (Supplementary Data 1 and 3). Our finding of close to equal numbers of gains of RNS in the ancestors of species-rich and species-poor groups supports the previous conclusion that the evolution of nodulation is not directly associated with increased speciation rates[56] although more work is needed. Similarly, uncertainty remains regarding the ecological contexts responsible for promoting the gain and maintenance of RNS, with current hypotheses focusing primarily on historical atmospheric conditions[11,57] and soil environments[58]. Confident placements of RNS gains on a phylogenetic tree also provide a means to date the gains of RNS and thus better test alternative hypotheses regarding the extrinsic, historical conditions and intrinsic factors that may have enabled gains and losses of the symbiosis trait[57].

## Considering the alternative: nodulation gain in the ancestor of the NFC

Our extensive, highly curated, and well-sampled data drove model selection and character state reconstruction, definitively identifying the ancestor of the NFC as non-nodulating. However, some authors have cited the asserted inability of phylogenetic methods to account for external information about the relative difficulty of nodulation gain and loss as a rationale for discarding the conclusions of these methods[11], despite the fact that modeling transition rates is directly related to modeling ease of trait gain and loss. This perspective, in particular regard to RNS, is partly motivated by a recently identified RNS-related gene presence/absence pattern in the NFC[35], where many losses of RNS were proposed as necessary to explain the many absences of two critical nodulation-associated genes, *NIN* and *RPG*, in non-nodulating species. This explanation implicitly presumes both simultaneous gain and coelimination[59,60] between RNS genes and the RNS trait. However, under a multiple-gains hypothesis, essential RNS-related genes could be recruited without an RNS phenotype and then become dispensable and thus fragmented or lost in lineages that do not also eventually gain RNS. This latter scenario is supported by the fact that not all non-nodulating species lack these key RNS genes[35].

To address the concern that phylogenetic methods cannot directly account for the different weights of gain or loss of RNS suggested by genomic information, we tested evolutionary models with a range of gain and loss rates to see under which relative rates the ancestor of the NFC is more likely to be nodulating than non-nodulating, consistent with the single-gain multiple-loss hypothesis proposed to explain gene presence/absence[35]. Even at the upper limit of the rate of loss tested (when loss of nodulation is over 60 times more likely than gain), a marginal character state reconstruction estimates that the ancestor of the NFC lacks RNS (Supplementary Fig. 8). An analysis in which the ancestor of the NFC is fixed as nodulating, thus requiring the gain proposed by the single-gain multiple-loss hypothesis[35], shows that an extremely high rate of loss is required to accommodate this ancestral state (Supplementary Fig. 9); this high-loss rate model is a dramatically poorer fit to the data than the best model (Supplementary Table 2).

Furthermore, we reconstructed the actual history of gains and losses of RNS implied by the unsupported scenario constraining the ancestor of the NFC as nodulating[35]. We found that the most likely scenario is one in which ancestral RNS is rapidly lost to the precursor state twice (once in the ancestor of Fagales/Fabales and once in the ancestor of Rosales/Cucurbitales) and is then regained through a history of gains identical to those in the best, unconstrained model of RNS gain and loss (Supplementary Fig. 9). Thus, forcing the estimation of a rapid loss rate through fixing the ancestral state still did not facilitate the reconstruction of a single gain and many losses. We consider this

result additional compelling evidence that even a biological predisposition to lose RNS quickly is still highly unlikely to have led to a one-gain, massive-loss scenario. Therefore, our results are robust to a variety of interpretations of RNS evolution and are not sensitive to a particular model or evolutionary rate estimate; even forcing a recently invoked scenario of multiple losses[35] cannot prevent the reconstruction of numerous gains.

Our finding of a two-step process of an ancestral precursor state giving rise to a labile intermediary hidden state from which RNS gain is likely supports a multi-level model of the predisposition hypothesis not considered previously. Our results suggest dispensing with simplistic views of the homology of RNS and the nodule organ; instead, we argue for limited homology among distinct lineages that gained nodules through independent intermediate states. We also identify with unprecedented resolution which species in the NFC may harbor genetic changes that precipitated the gain of RNS and therefore have significant implications for transferring RNS to candidate crop species.

Efforts to elucidate the molecular underpinnings of RNS have so far focused on model legumes[48]. Yet, to understand what genomic innovations underlie the multi-step model supported by our analyses, it is imperative to use a more diverse set of model organisms for deep functional studies. However, this much-needed approach is hampered by the lack of a genetic system for *Frankia*, the difficulty of obtaining actinorhizal plant mutants, and the slower development of most actinorhizal plant species[61]. Although efforts to understand the *Frankia*/actinorhizal root nodule symbiosis lag behind the study of rhizobial symbiosis in legumes[61], these developing efforts are more evenly distributed across independent gains of RNS in non-legumes than is the case within the legumes; every independent gain of RNS outside of legumes is represented by an established or nascent research program except the gain in Myricaceae[62,63]. Similarly, studies of species that appear to represent independent losses of RNS, as inferred here, will be vital to reconstruct the processes that led to RNS loss. Further study of lineage-specific nodulation traits in non-legumes[61], as well as broader representation within the legumes, will be essential for further tests of our hypothesis of independent multi-step gains of RNS and, ultimately, for enhanced use of these results in engineering nitrogen-fixing symbioses in candidate crops.

## Methods
### Specimen sampling and data generation
Our sampling of species used herbarium specimens exclusively; we set a series of sampling goals to broadly represent taxa in the NFC, achieving near-complete sampling of recognized genera with close to proportional sampling of each genus while ensuring that areas of the phylogeny with high variation in the nodulation trait were particularly well represented. The phylogenetic tree presented here was built from genomic data from 12,768 herbarium specimens. Further details of the sampling methodology and sample management workflow have been reported previously[26], and a detailed breakdown of sampling at ordinal, familial, and generic levels may be found in Supplementary Table 1.

DNA extraction was performed using a modified CTAB protocol in a high-throughput 96-well format that has been described previously[26]. DNA extracts were quantified via PicoGreen, and samples with total DNA amounts below 10 ng were excluded from downstream sequencing. Standard genomic library preparation was performed but omitted a sonication step because DNA derived from herbarium specimens is typically in the size range required for sequence capture and Illumina sequencing.

We designed a panel of 100 loci for phylogenomic analysis using MarkerMiner[64], using *Arabidopsis thaliana* (rosids: Brassicales) as the genomic reference and 78 representative transcriptomes across the rosid clade derived from the 1KP project[65,66]. Beyond the filtering criteria implemented in MarkerMiner, we required loci to be at least 500 bp long with at least 50% species coverage of the 78

transcriptomes to ensure the probe set's applicability clade-wide. Ultimately, probes were designed covering loci redundantly with 20 phylogenetically representative species to span sequence variation in the NFC; the overall capture space was 133,433 bp (aligned length), covered by 22,749 probes (100 bp; 2× tiling) with a mean pairwise identity of 72.2% among phylogenetic representative sequences. These probes were used for standard multiplex sequence capture and Illumina HiSeq sequencing performed by Rapid Genomics (Gainesville, FL, USA).

Raw reads were trimmed and quality filtered using Trimmomatic v. 0.39[67] to scan with a 20-base sliding window and cut when the average quality per base dropped below 15. We used FastQC[68] to assess quality post-trimming. To screen for and remove contaminated samples, we used aTRAM v. 2.0[69] to assemble the ITS locus from each sample. We then used ITS sequences to query against the NCBI GenBank database to check for erroneous taxonomic identity.

Target loci were assembled using a custom single-copy filtering pipeline based on Yang and Smith (2014)[70] and aTRAM[8] (Supplementary Note 2). Within this pipeline, de novo assembly was conducted with SPAdes[70,71] using a coverage cutoff of 40× and a length threshold of 500 bp. Efforts to reduce the analysis of paralogous sequences are crucial to robust phylogenetic inference[70]. Due to the size and phylogenetic diversity of our dataset, the best strategy for reducing paralogs was to remove loci with paralogs for a given sample. We used a gene tree approach to drop any sample for which more than one distinct, high-quality contig was assembled (Supplementary Note 2). By only dropping tips from particular loci (in cases where multiple contigs were assembled, indicating possible paralogy issues), we were able to include representation of nearly all sampled/sequenced species across most of loci examined. We removed 14 of 100 loci entirely due to high paralogy across samples. Our conservative approach to selecting orthologs also provides a safeguard in cases where allopolyploidy might obfuscate the signal of the species tree. Pseudo-orthology due to diploidization should be problematic only under very specialized conditions not likely to obtain here[72].

## Phylogenetic analyses

To overcome computational and mathematical limitations[73], we did not estimate a total-evidence phylogeny for our entire dataset. Instead, we used a subtree scaffolding approach in which a backbone phylogeny was estimated using both taxon and nucleotide resampling. To perform taxon resampling, we randomly subsampled 500 taxa that proportionally represented each of 15 major subclades (Fagales; Cucurbitales; Rosaceae, Rhamnaceae, Elaeagnaceae, Ulmaceae, Cannabaceae, Moraceae, and Urticaceae or Rosales; Polygalaceae of Fabales, Papilionoideae, Caesalpinioideae, Cercidoideae, Detarioideae, Dialioideae of Leguminosae, and outgroups; for major subclades with few samples (e.g. the Leguminoisae subfamily Duparquetioideae), all samples were added to each sample set). For each of 20 subsampled taxon sets, we aligned assembled sequences using MAFFT v7.294b with high accuracy options[74]. For backbone-tree estimation, we (1) performed 1,000 rapid bootstrap replicates in a concatenation and maximum-likelihood (CA-ML) analysis using IQ-tree[75] (Supplementary Fig. 10; p 11–20) and (2) estimated separate gene trees for all loci in IQ-tree for species-tree estimation with ASTRAL-III[76] (Supplementary Fig. 10; p. 1-10). Because not all subsets attained strong backbone support, we used the species-tree topology with the best global support values (Supplementary Fig. 10; p. 3) as a scaffold for arranging independently estimated subclade phylogenies.

Estimation of subclade phylogenies included all samples for each of 15 subclades (see above) as well as 15 outgroups sampled from other subclades. To estimate these subclade phylogenies, we aligned sequences using MAFFT v7.294b with high-speed options and reconstructed phylogenies from concatenated data matrices using CA-ML in RAxML-NG[77]. We did not explore species-tree estimation for the subclades due to limited information in individual gene trees with many taxa (5121 taxa in the largest subclade, Papilionoideae) and short alignments[78,79], and we did not perform bootstrapping on the subclade phylogenies because of computational limitations and methodological concerns regarding the utility of Felsenstein's bootstrap for large phylogenies[80]. For outgroups and major subclades with few samples, samples were added to a closely related subclade for sequence assembly and omitted from phylogeny reconstruction. We added these samples to the total phylogeny based on their position in the backbone phylogeny described above.

We combined subclade phylogenies manually using the backbone phylogeny described above. Branch lengths on this phylogeny were estimated from a total alignment of all samples across all subclades using the command raxml-ng – estimate in RAxML-NG. After that, we performed time-calibration using treePL v.1.087[81] with 12 calibration points based on Sun et al.[30] covering all four orders and eight of 15 families in the NFC. Using the dates for the NFC from Sun et al.[30,81] allowed us to directly integrate our time-calibrated NFC phylogeny into the broader context of rosids. Although the evolution of RNS may be investigated at various phylogenetic scopes, we chose the broader context of the rosid clade to allow for inferences about shifts in transition rates in the NFC relative to the rest of the rosids without introducing so much rate heterogeneity that the signal within the NFC might be obfuscated.

## Compiling accurate databases for taxonomy and RNS

Our internal taxonomic reconciliation, used both to guide the sampling and to associate phylogenetic and trait data, followed The Plant List v. 1.1. For legumes, the taxonomy was updated before trait-scoring using an unpublished database provided by the Royal Botanic Gardens, Kew, and the Legume Phylogeny Working Group to eliminate cases of clear non-monophyly.

We scored tips in our phylogeny for the presence or absence of RNS using a significantly expanded RNS database (Supplementary Data 2). To compile our database of RNS status at the genus level, we first digitized data from the two most comprehensive texts on legume nodulation[82,83]. Although other published resources may conflict with or supplement these sources[56,84], there is uncertainty regarding single reports of nodulation in the field without laboratory confirmation that may instead represent morphological misinterpretations[85], motivating the high standards of evidence used here. For non-legumes, scoring presence/absence of nodulation is more straightforward because of both fewer species and a lack of new observations of nodulating species in recent decades[86]. For both legumes and non-legumes, we confirmed and updated published reports with unpublished data from field experts (Supplementary Note 3).

Despite attempts to apply high standards described above, any strategy to score RNS comprehensively at a broad scale is imperfect. First, observation of nodulation competency is not straightforward since many nodulators can facultatively produce nodules, particularly in tropical environments[87]. Second, gross morphology is not sufficient to ascertain the presence of nodules, and there are published reports of nodules that likely represent disease structures[83]. Finally, the diversity of nodule structures[83,88] and basic questions about the homology of nodules[20] highlight the importance of understanding not only what nodulators share but what distinguishes them; a global approach treating all nodulators as equivalent has the potential to anonymize potentially diverse trait histories. Hence, our understanding of what species and clades are nodulating is only as good as the reports we have from those engaged in the time-intensive task of identifying, confirming, and characterizing nodulation; even at its best, an attempt to characterize nodulation across the entire clade yields uncertainty.

We sought to maximize information and minimize the effects of errors on our overall results by using a strategy of RNS status informed

by phylogeny. Given the limitations described above, we used the following approach to capture nodulation competency in scoring genera. Genus-level RNS status determinations are usually based on observations of nodulation in one or a few species[14,56]. We extended this approach to our method for scoring genera for which no nodulation data are available by assigning the state of their closest relatives in those areas of the tree where RNS states are uniform. RNS states are uniform in many large clades with little missing data; scoring was straightforward in all but a few genera occurring in hotspots of RNS evolution, and these genera were scored as unknown (Supplementary Data 2, Supplementary Note 3). After improving the taxonomy as described above, few genera remained non-monophyletic, and in most cases, we were able to score species of polyphyletic or paraphyletic groups because they occurred in areas of the tree with uniform RNS status (Supplementary Data 2, but see Supplementary Note 3).

### Modeling the evolution of RNS

Simple models of binary character evolution, while appropriate for small clades, are not likely to adequately explain the evolution of binary characters in larger, older, and globally distributed clades where the imposition of a homogeneous evolutionary model is not reasonable because the lability of a trait is nearly certain to differ among clades[52]. Erroneously constraining evolutionary rates across broad spans of time and phylogeny can lead to incorrect reconstruction of ancestral character states[52]. We used hidden state models as implemented in the R package corHMM[52] to identify different rates of evolution of RNS along branches of the rosid phylogeny.

We performed model selection on models that included one to five hidden states, placing no constraints on the rate of gain and loss of RNS (i.e., unequal rates models). We did not test additional numbers of hidden states due to computational constraints and a dramatic increase in AIC values between three and four rate classes, suggesting overparameterization (Supplementary Table 2). For all models with more than one hidden state, we also tested a model in which gains of RNS cannot occur in one of the hidden states based on the prior knowledge that RNS gain does not occur outside the NFC (Supplementary Table 2). We ran 100 random restarts for each model. For our best-fitting model, we estimated uncertainty for all parameter estimates by sampling points around Δ2 from the maximum likelihood estimates using the R package dentist (https://github.com/bomeara/dentist; Supplementary Table 3) – the advantage of this approach is that it allows the detection of covarying parameter values that may form a ridge in likelihood space. We then performed a joint estimation of ancestral character states using transition rates inferred under the best-fitting model as implemented in corHMM. Results of ancestral character state reconstruction mapped onto the rosid phylogeny were visualized and plotted using the R package "ggtree" release 3.4.1[89]. We visualized results of transition rate estimation with the R package "RateViz" (https://github.com/bomeara/RateViz).

Estimation of the number of character state transitions was based on comparing the estimates of states at the beginning (parent) and end (child) node of each edge of the tree. A transition was noted every time the most likely estimate at the two nodes differed in the observed state. As this is the joint reconstruction, the states at nodes represent the single likeliest reconstruction for all nodes at once. Thus, a difference in state between branch start and end implies that there is at least one change on that branch. However, reconstructing changes in this way could underestimate change: it is possible, for example, for a state to change and then change back on a long edge, and this would not be detectable using our approach. While approaches more sensitive to reversed changes, such as stochastic character mapping[90], could be used instead, our approach should nevertheless effectively provide reliable (and conservative) estimates of the minimum overall number of changes.

In addition to transition rate estimation and ancestral character state reconstruction under the best-fit model as determined by rigorous model selection, we performed transition rate estimation and ancestral character state reconstruction under three additional models: 1) The two-rate precursor model (ΔAIC 1.063; Supplementary Fig. 11); 2) A fixed-ancestral-state analysis with RNS-present (ΔAIC 6.14) to represent the single-gain, multiple-loss model of Griesmann et al.[35] (Supplementary Fig. 9); and 3) A two-rate, three-state custom model (three states: absent, actinorhizal-RNS, rhizobial-RNS; Supplementary Fig. 12) (ΔAIC 49.176) to consider the scenario where actinorhizal RNS can be gained from a precursor state and rhizobial RNS can be gained from the actinorhizal state. This scenario is based on the hypothesis that actinorhizal-type nodules are ancestral and that legume-type nodules are derived[91].

### Robustness to phylogenetic uncertainty

A significant source of uncertainty in trait reconstruction is underlying phylogenetic uncertainty in the NFC. Because of the size and complexity of our dataset, it was not feasible to conduct any tests that involved iterating over many different topologies with various permutations of within-clade relationships, such as bootstrap replicates. We focused instead on the impact of different topologies in two of the most notoriously uncertain areas of the NFC: the relationships among clades recognized taxonomically as orders (Fabales, Rosales, Cucurbitales, Fagales)[16] and the relationships among legume subfamilies sensu LPWG (2017)[92]. For these tests, we used the NFC backbone resolved by Zanne et al.[32] following[16] and the more recent Leguminosae backbone phylogeny resolved by Koenen et al.[28] to construct a supertree from our independently analyzed subclades (relationships among families in Rosales remained the same as in our primary analysis) (Supplementary Note 4) and performed hidden rate modeling and ancestral character state reconstruction following the same steps described for the main topology.

### Comparing the likelihood of ancestral presence vs. absence

To generate a list of RNS-gain and RNS-loss rates to consider, we first ran a model with no hidden states on a character matrix and phylogeny for which the NFC ancestor was not fixed as RNS-present or RNS-absent. We generated an even distribution of 100 possible gain and loss rates that included the most-likely estimate of gain and loss rates in the model and ran a marginal character state reconstruction for each of these gain-and-loss rate pairs. For each analysis, we plotted the log-likelihood of the result and how likely the ancestor of the NFC was to possess or lack RNS.

### Reporting summary

Further information on research design is available in the Nature Portfolio Reporting Summary linked to this article.

## Data availability

Supporting Information is available online in Supplemental Information and Data, including RNS state database, phylogenetic trees, and detailed plots of ancestral reconstruction. Sequence data are available on SRA (BioProject PRJNA1021556 (https://www.ncbi.nlm.nih.gov/bioproject/1021556), PRJNA1021608 (https://www.ncbi.nlm.nih.gov/bioproject/1021608), PRJNA1021620 (https://www.ncbi.nlm.nih.gov/bioproject/1021620), PRJNA1022015 (https://www.ncbi.nlm.nih.gov/bioproject/1022015), PRJNA1022023 (https://www.ncbi.nlm.nih.gov/bioproject/1022023), PRJNA1022025 (https://www.ncbi.nlm.nih.gov/bioproject/1022025), PRJNA1022027 (https://www.ncbi.nlm.nih.gov/bioproject/1022027), PRJNA1022029 (https://www.ncbi.nlm.nih.gov/bioproject/1022029), PRJNA1022030 (https://www.ncbi.nlm.nih.gov/bioproject/1022030), PRJNA1022032 (https://www.ncbi.nlm.nih.gov/bioproject/1022032), PRJNA1022323 (https://www.ncbi.nlm.nih.gov/bioproject/1022323), PRJNA1022138 (https://www.ncbi.nlm.nih.

gov/bioproject/1022138), PRJNA1022141 (https://www.ncbi.nlm.nih.
gov/bioproject/1022141), PRJNA1022147 (https://www.ncbi.nlm.nih.
gov/bioproject/1022147)), and the phylogenetic tree from Fig. 1 and
Supplementary Fig. 1 is available in OpenTree (Study ID ot_2291;
https://tree.opentreeoflife.org/curator/study/view/ot_2291) and in
newick format as Supplementary Data 3. A concatenated alignment file
including all samples (used for branch length estimation and can be
parsed into smaller subsets for subtree analyses) is at 10.5281/
zenodo.10728230.

## Code availability

Code to reproduce ancestral reconstruction analyses are posted on
GitHub: https://github.com/HeatherKates/Kates2024NatureComm.

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

## Acknowledgements

This work was supported by DOE grant DE-SC0018247 to M.K., R.G., P.S., and D.S. and a UFBI grant (University of Florida). We thank Katharina Pawlowski for reviewing our scoring of actinorhizal symbiosis and for related discussions. We thank Colin Hughes and other members of the Legume Phylogeny Working Group for reviewing and helping to resolve taxonomy issues in Leguminosae. We thank Mark Whitten, Kelly Balmant, Chris Dervinis, Joshua Dieringer, and Henry Schmidt for help with specimen sampling. Tingshuang Yi acknowledges funding from the National Science Foundation of China (No. 31720103903).

## Author contributions

P.S., D.S., R.G., R.F., M.K., and H.K. conceived of the study. R.G., S.L., Q.T., R.F., D.C., H.K., P.S., T.Y., and D.S. collected herbarium specimens. R.F. designed probes for target-enrichment. H.K., Q.T., and S.L. extracted DNA. H.K. assembled sequence data and performed phylogenetic analyses. G.S. and H.K. designed the single-copy-filtering pipeline. R.L. built and managed specimen metadata databases. B.O., J.M., and H.K. compiled hidden rate models to test via model selection. B.O. wrote the code to simulate transition rates for a two-rate model of RNS gain and loss. E.J. helped to review and made novel contributions to the RNS trait database. All authors contributed to the final manuscript.

## Competing interests

The authors declare no competing interests.
