## [Peer Review File · Nature Communications]

Shifts in evolutionary lability underlie independent gains and losses of root-nodule symbiosis in a single clade of plantsEditorial Note: This manuscript has been previously reviewed at another journal that is not operating a transparent peer review scheme. This document only contains reviewer comments and rebuttal letters for versions considered at *Nature Communications*.

Reviewer #1 (Remarks to the Author):

I was referee 1, and I am seeing the new manuscript and response from the referees.

I reiterate my initial comments that the large size of tree produced here is impressive, and the question about how many times root-nodule symbioses (RNS) have evolved in angiosperms is important. I acknowledge the fact that several contradictory accounts have been published so far.

Because less than half of the species in the nitrogen fixing clade have been sampled, I had suggested to place missing ones with taxonomies in a more complete tree. The authors explain that they had scored NDS at genus level and therefore decided not to implement a species-level analysis. I think this limits the paper – are we then just seeing ‘an intermediate stage’ and need to await a larger tree to get the ‘true’ number of gains and losses of RNS? The authors could have taken a subclade where to add all species to give us some idea of how the number of gains/losses will be inflated once we have the full picture (i.e. a species level tree).

I also noted some discrepancies in explaining the origins of RNS. The paper discusses intrinsic factors (speciation rates) and cites potential extrinsic factors (Sci Rep 2015) without comprehensive discussion. I did not suggest they test extrinsic factors, but an expanded discussion here of all the factors involved, whether extra or intrinsic, would be most useful. The added text cited in the response does not do the job well. Interestingly, the authors also explain they are working on extrinsic factors but for another paper.

I still cannot see a data availability statement. Where is the tree file (not the tree image) freely available? It should be in TreeBase.

I am not satisfied with the authors’ response with regard to scoring nodulation. I know it is difficult, but they should provide a comprehensive data set where we can trace back their decisions. A large table indicating for each genus with species was used as exemplar and where the data came from should be available as part of this paper.

Reviewer #2 (Remarks to the Author):

Having served as a reviewer for a previous version of this manuscript in 2022 (to another journal in the Nature group), I am satisfied by the revision and the authors’ responses to comments from myself and other reviewers. This is an impressive synthesis of a great deal of phylogenetic data and trait information. I think it makes a strong case for the proposed model of multiple RNS gains and rare losses. The question of underlying mechanisms remain tantalizingly absent, but I think the present results stand on their own.

Manuscript ID: NCOMMS-23-04938A

Response to Reviewers

08/17/2023

REVIEWER 1

I reiterate my initial comments that the large size of tree produced here is impressive, and the question about how many times root-nodule symbioses (RNS) have evolved in angiosperms is important. I acknowledge the fact that several contradictory accounts have been published so far.

We thank the reviewer for noting the important novel aspects of our study.

Because less than half of the species in the nitrogen fixing clade have been sampled, I had suggested to place missing ones with taxonomies in a more complete tree. The authors explain that they had scored NDS at genus level and therefore decided not to implement a species-level analysis. I think this limits the paper – are we then just seeing ‘an intermediate stage’ and need to await a larger tree to get the ‘true’ number of gains and losses of RNS? The authors could have taken a subclade where to add all species to give us some idea of how the number of gains/losses will be inflated once we have the full picture (i.e. a species level tree).

We have followed the senior editor’s suggestions as noted above. We have not implemented the specific suggestion, but we have taken to heart the more central intellectual point being made. We have added text in a new paragraph placed in the section entitled, “Precise phylogenetic origins of RNS”. There we identify strengths and limitations of our estimate, focusing on the central point raised above that increased phylogenetic data are less likely to generate new insights than new nodulation data.

I also noted some discrepancies in explaining the origins of RNS. The paper discusses intrinsic factors (speciation rates) and cites potential extrinsic factors (Sci Rep 2015) without comprehensive discussion. I did not suggest they test extrinsic factors, but an expanded discussion here of all the factors involved, whether extra or intrinsic, would be most useful. The added text cited in the response does not do the job well. Interestingly, the authors also explain they are working on extrinsic factors but for another paper.

We have gone over this cited section, which was too brief in the original submission as noted, for clarity and expanded discussion of potential factors. As the reviewer suggests, we have focused on a literature review and presentation of existing hypotheses, focusing on soil nutrient limitation and historical CO₂ concentration in the atmosphere, both discussed in seminal papers by Sprent as cited. This section has been reframed to focus on outstanding questions, because the state of the literature actually indicates that we have little definitive knowledge on those extrinsic factors, a situation partly due to shortcomings in previous phylogenetic and nodule origin hypotheses as addressed in the manuscript, but the framework included in this paper will

be an important foundation to test alternative hypotheses regarding what promotes nodulating strategies in ecological communities.

I still cannot see a data availability statement. Where is the tree file (not the tree image) freely available? It should be in TreeBase.

In accordance with *Nature* Portfolio requirements, which we have reviewed, the data availability statement is now provided. The tree file as requested has been made available to the reviewer and will be uploaded to data dryad upon acceptance in accordance with *Nature* Portfolio policies.

I am not satisfied with the authors' response with regard to scoring nodulation. I know it is difficult, but they should provide a comprehensive data set where we can trace back their decisions. A large table indicating for each genus with species was used as exemplar and where the data came from should be available as part of this paper.

We have provided an updated version of Supplementary Table 1 which includes tracking of literature sources underlying the nodulation determinations for all genera. Because we did not use species-level RNS data in our study, it is beyond the scope of our database to include this information in our supplementary information; moreover, there is no species-level information for the vast majority of genera (which are RNS-) in Supplementary Table 1. Readers may follow the literature cited in Supplementary Table 1 to find this information.

REVIEWER 2

Having served as a reviewer for a previous version of this manuscript in 2022 (to another journal in the Nature group), I am satisfied by the revision and the authors' responses to comments from myself and other reviewers. This is an impressive synthesis of a great deal of phylogenetic data and trait information. I think it makes a strong case for the proposed model of multiple RNS gains and rare losses. The question of underlying mechanisms remain tantalizingly absent, but I think the present results stand on their own.

We thank the reviewer for evaluation of the strengths of our work. We acknowledge that the mechanistic perspective is a key unknown, but the underlying mechanistic work was beyond the scope of this already massive phylogenetic undertaking. Critically, the firm phylogenetic foundation we provide here will be an important framework for future studies that need detailed information on phylogenetic distributions and ancestral origins in order to make better choices of species for comparison to facilitate the identification of mechanisms.

Reviewer #1 (Remarks to the Author):

Some progress has been achieved, but there are still important gaps in the information provided.

I appreciate the inclusion of the paragraph detailing the precise phylogenetic origins of RNS.

I thoroughly reviewed the Data Availability section and attempted to access the provided links. Unfortunately, I encountered an 'Access Denied' message when trying to open the Treebase link. Furthermore, I couldn't locate the Bio Project PRJNA986853 in GenBank's database as of now. It's worth noting that there is no mention of Dryad, contrary to what was stated in the rebuttal.

Moving on to the Code Availability, I attempted to access the GitHub link. However, I found that the Readme lacks explanatory text, and some of the included files appear unrelated to the present article's subject (such as those related to dentist analyses). This situation indicates that both the data and the code are currently inaccessible to both readers and reviewers.

I'm pleased to acknowledge that Table S1 now offers some valuable references.

REVIEWERS' COMMENTS

Reviewer #1 (Remarks to the Author):

Some progress has been achieved, but there are still important gaps in the information provided.

I appreciate the inclusion of the paragraph detailing the precise phylogenetic origins of RNS.

I thoroughly reviewed the Data Availability section and attempted to access the provided links.

Unfortunately, I encountered an 'Access Denied' message when trying to open the Treebase link.

The Data Availability statement was provisional at the time of the last revision, and all relevant data have been made available in this submission and the updated Data Availability statement thoroughly details this. Regarding Treebase, we have instead deposited the tree on OpenTree (Study ID ot_2291; https://tree.opentreeoflife.org/curator/study/view/ot_2291); the references in the manuscript have been updated accordingly to refer to this.

Furthermore, I couldn't locate the Bio Project PRJNA986853 in GenBank's database as of now.

The SRA database (not GenBank, which does not accept raw short read data) has now fully released the data associated with this study. The project numbers are now different from the BioProject number noted above, and have been updated in the manuscript's Data Availability Statement. The PRJN numbers are now: PRJNA1021556, PRJNA1021608, PRJNA1021620, PRJNA1022015, PRJNA1022023, PRJNA1022025, PRJNA1022027, PRJNA1022029, PRJNA1022030, PRJNA1022032, PRJNA1022323, PRJNA1022138, PRJNA1022141, and PRJNA1022147.

It's worth noting that there is no mention of Dryad, contrary to what was stated in the rebuttal.

We have decided, now that the data deposition is complete, to keep all of the analysis data and code in one place on GitHub (<https://github.com/HeatherKates/Kates2024NatureComm>). The Data Availability Statement has been edited to reflect this.

Moving on to the Code Availability, I attempted to access the GitHub link. However, I found that the Readme lacks explanatory text, and some of the included files appear unrelated to the present article's subject (such as those related to dentist analyses). This situation indicates that both the data and the code are currently inaccessible to both readers and reviewers.

The GitHub for the project is now public (<https://github.com/HeatherKates/Kates2024NatureComm>) and has implemented more documentation and a detailed README.md file. The Github repository does still includes the “Dentist” analysis, which is the name of an R package that is indeed referenced in the main text, which pertains to estimating modeling uncertainty.

I'm pleased to acknowledge that Table S1 now offers some valuable references.

Thank you, and we appreciate the previous reviewer comment requesting these references, which has improved the value of Table S1 (now Supplementary Data 2).